# Preliminary Results of Marine Gravity Recovery by Tiangong-2 Interferometric Imaging Radar Altimeter

Meng Sun [1,2], Yunhua Zhang [1,2,*], Xiao Dong [1,2] and Xiaojin Shi [1,2]

1 CAS Key Laboratory of Microwave Remote Sensing, National Space Science Center, Chinese Academy of Sciences, Beijing 100190, China; sunmeng18@mails.ucas.ac.cn (M.S.); dongxiao@mirslab.cn (X.D.); shixiaojin@mirslab.cn (X.S.)
2 School of Electronic, Electrical and Communication Engineering, University of Chinese Academy of Sciences, Beijing 100049, China
* Correspondence: zhangyunhua@mirslab.cn

**Abstract:** This paper presents for the first time the results of marine gravity recovery using the ocean observation data acquired by Tiangong-2 interferometric imaging radar altimeter (TG2 InIRA) which demonstrate not only the balanced accuracies of the north and east components of deflection of the vertical (DOV) as envisaged, but also the improved spatial resolutions of DOV compared with that by conventional altimeters (CAs). Moreover, much higher measurement efficiency owing to the wide-swath capability and the great potential in accuracy improvement of marine gravity field are also demonstrated. TG2 InIRA adopts the interferometry with short baseline and takes small incidence angles, by which wide-swath sea surface height (SSH) can be measured with high accuracy. Gravity recovery experiments in the Western Pacific area are conducted to demonstrate the performance, advantages and capability of TG2 InIRA. SSH data processing algorithms and DOV calculation have been designed by taking the wide-swath feature into account, based on which, the gravity anomalies are then calculated using the inverse Vening Meinesz formula. The derived gravity anomalies are compared with both the published gravity models and the shipborne gravity measurements. The results show that the accuracy of TG2 InIRA is equivalent to, or even a little better than, that of CAs. The fused gravity result using equal TG2 InIRA data and CAs data performs better than those using TG2 InIRA data alone or CAs data alone. Due to the signal bandwidth of TG2 InIRA is only 40 MHz which is much smaller than that of CAs, much higher accuracy can be hopefully achieved for future missions if larger signal bandwidth is used.

**Keywords:** Tiangong-2 interferometric imaging radar altimeter (TG2 InIRA); wide-swath altimeter (WSA); error correction; sea surface height (SSH); deflection of the vertical (DOV); marine gravity anomaly; gravity recovery

## 1. Introduction

Marine gravity data has lots of scientific applications, such as marine bathymetry [1,2], seafloor topography [3,4], tectonic plates [5,6] and lithospheric structure [7]. Moreover, it also has many engineering applications, such as correction of inertial navigation [8], exploration of marine resources [9] and planning of shipboard surveys [10]. The quality of marine gravity recovery relies on the accuracy and resolution of accumulated multi-satellite altimetry data, which are mainly decided by the following factors as commonly known: (1) altimeter range precision, which is related to the altimeter system characteristics [11]; (2) spatial coverage, which is related to mission duration and resampling frequency [12]; (3) orbital track orientation related to orbital inclination and latitude [13]; and (4) pulsed footprint of nadir altimeter. Beginning with the early GeoSat and ERS-1/2 missions to the later Jason-1/2/3, SARAL and CryoSat-2 missions, satellite altimeters have continuously collected a huge amount of sea surface height (SSH) observation data for decades, and

many scholars have developed different gravity recovery methods along with lots of experiments, such as the Laplace equation method [10,14–16], the inverse Vening-Meinesz (IVM) method [17–20], the inverse Stokes integral method [21–23] and the least-squares collocation method [24–26]. The advancements of satellite altimeter and recovery methods make the quality of gravity field recovery continuously improved.

However, due to the pulse-limited delay mode that is adopted by conventional altimeters (CAs) to measure the SSH, there exist obvious limitations on further improving the accuracy and resolution of SSH measurement, which result from the large footprint [11,27]. In addition, the CA can only measure one-dimensional SSH profile along the track but lack measurements across track [28]. When a satellite altimeter operates in large inclination orbits, especially for the polar orbit, the inconsistent accuracy of the north and east components of deflection of the vertical (DOV) is unavoidable, and the subsequent gravity recovery can thus be seriously affected [13,29].

These problems may soon be alleviated with the new generation wide-swath altimeter (WSA), such as the Ka-band radar interferometer (KaRIn) of the Surface Water and Ocean Topography (SWOT) mission [30,31]. The WSA, also known as interferometric imaging radar altimeter (InIRA), utilizes interferometry to measure SSH unlike CA uses echo waveform under nadir looking [32,33]. The accuracy of SSH mainly depends on the measurement accuracy of interferometric phase [34,35]. The application of interferometry makes InIRA capable of measuring SSH with wide swath (e.g., tenths of kilometers in each side of track); thus, the observation efficiency can be greatly improved [31]. The adopted small incidence angle helps for obtaining high signal to noise ratio (SNR) which is the precondition for high accuracy SSH measurement via interferometry. In addition, compared with the CA, the wide-swath characteristic of InIRA helps for improving the accuracy of the east component of DOV [28,29]. The consistency of the north and east DOV accuracies is beneficial to subsequent gravity recovery [13]. It is foreseeable that the SSH observation data from InIRA will derive an unprecedented high-quality gravity filed [36]. Some scholars have carried out the gravity recovery experiment based on simulation data. Gaultier et al. (2016) have developed a SWOT simulator to simulate SSH measurements [37]. Jin et al. (2022) and Yu et al. (2021) have utilized the SWOT simulator to generate SSH data, and based on which, the DOV and gravity anomaly are calculated to show the advancement of the SWOT mission [29,38].

On 15 September 2016, the Chinese Tiangong-2 space laboratory was launched onboard with an interferometric imaging radar altimeter (TG2 InIRA), which is the first spaceborne WSA in the world. TG2 InIRA aims to validate the working principle and payload design for wide-swath SSH measurement. It adopts small incidence angle interferometry, aperture synthesis and ocean-land compatible height tracking technologies to realize wide-swath SSH measurement and three-dimensional topography reconstruction [39]. Benefiting from these new technologies, TG2 InIRA has shown great potential in both ocean observation and terrestrial water measurement. In the past few years, some studies have been conducted based on the observation data of TG2 InIRA, such as significant wave height (SWH) retrieval [40], wind speed retrieval [41,42] and joint retrieval of wind speed and SWH [43], internal wave detection [44] and river-lake water level measurement [45]. In our previous work [46], we presented a very initial result of DOV using the TG2 InIRA SSH data without baseline correction and noise suppression. In this paper, we present the preliminary gravity recovery results from TG2 InIRA data for the first time.

The value of the paper lies in realizing the gravity recovery from InIRA observation data for the first time, and the achieved preliminary results show the great potential of InIRA in SSH measurement and gravity field recovery. In Section 2, we introduce the processing of the TG2 InIRA raw data, including synthetic aperture imaging, interferometric phase processing, and correction of errors. In Section 3, observation data in a Western Pacific region specified by 139°E–146°E and 21°N–28°N are collected from near three-year observation data. Then, the preprocessing algorithms are designed and applied to the TG2 InIRA SSH data. Section 4 describes the calculation methods of DOV and gravity anomalies

with the wide-swath characteristics taken into consideration. Section 5 presents the gravity recovery results which are compared to the published models and the shipborne gravity measurements. Conclusions are finally drawn with the outlook in Section 6.

## 2. SSH by TG2 InIRA

### 2.1. TG2 InIRA

The Chinese Tiangong-2 space laboratory was boarded with several payloads and the InIRA is one of them. The orbit altitude of the Tiangong-2 space laboratory is around 393 km, and the inclination is about 43°. The TG2 InIRA worked at Ku-band (13.58 GHz) with 40 MHz bandwidth, whose interferometric baseline is 2.3 m long and 5° inclined referring to the horizontal. The data from 2.5 to 8° incidence angles are used covering about 35 km single right swath at normal conditions although the real incidence angle range is 1–10°. The original resolutions of SSH obtained by TG2 InIRA are about 30 m along-track, and 30–200 m across-track. The relative SSH accuracy of TG2 InIRA is about 0.1–0.3 m at 2 km × 2 km. Relative SSH accuracy refers to the uncertainty of relative height within the swath caused by random errors, which can be improved by spatial low-pass filtering of pixels and averaging the measurements of different passes.

The characteristics of small incidence angles and short baseline facilitate TG2 InIRA to obtain strong and high coherent reflections from ocean surface. Synthetic aperture processing and pulse compression technology are applied to improving the azimuth and range resolutions and in turn can provide much more independent measurements that is good for increasing the look-number to suppress the SSH noise. The sketch of the observation geometry of TG2 InIRA is presented in Figure 1 along with an in-orbit photo of TG2 spacecraft [39]; as shown, two antennas are installed beneath the resource module of TG2 spacecraft.

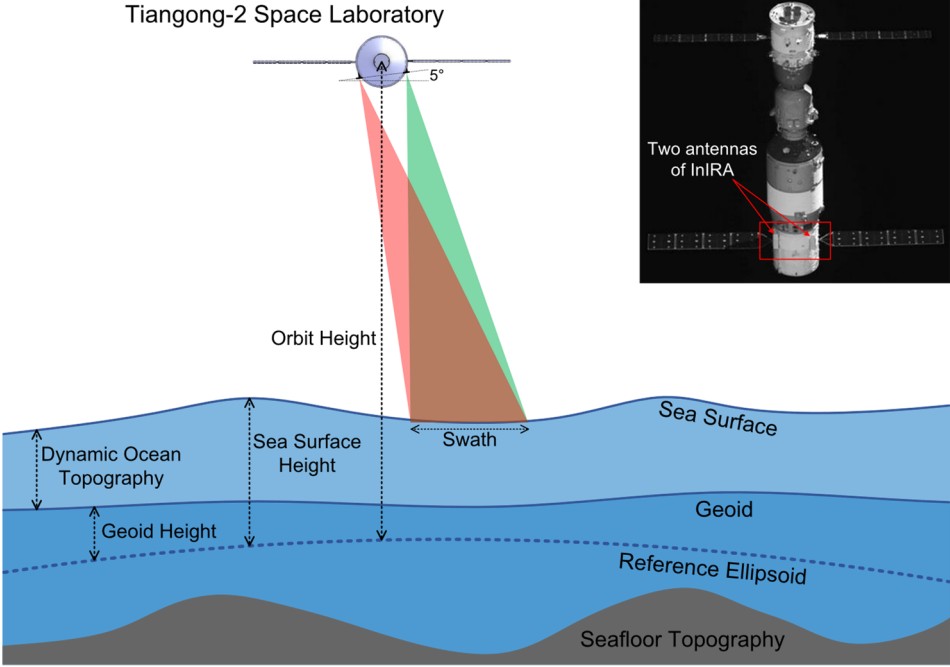

**Figure 1.** Observation geometry of TG2 InIRA for SSH measurement. Two antennas are installed directly beneath the resource module of TG2 spacecraft forming a 5° inclined baseline of 2.3 m long. The in-orbit photo of TG2 spacecraft was taken by an accompanying micro-satellite.

### 2.2. Synthetic Aperture Imaging and Interferometric Processing

The working principle and data processing method of the TG2 InIRA are almost the same as that of interferometric synthetic aperture radar (InSAR). TG2 InIRA forms the interferometry by directly installing two antennas with one transmitting signal while both

receive echo signal. The received raw data by two antennas are first processed by Range-Doppler synthetic aperture processing algorithm to form a pair of single-look complex (SLC) images, then the obtained interferometric phase by complex conjugation of the master and slave images is converted to the range difference referring to the same ground pixel, and finally, the height of every image pixel referring to a surface (here, it is the WGS84 ellipsoid surface) is obtained by solving the range-Doppler equation sets [47,48].

The imaging and interferometric processing is briefly summarized as follows. Range compression is first conducted by matching filtering using the calibration signal as the reference function, range cell migration compensation is then conducted in the range-Doppler domain, which is formed by Fourier transforming the range compressed signal with respect to the azimuth direction. SLC images are obtained after azimuth compression and used for further interferometric processing. Figure 2 presents the obtained interferometric phase image corresponding to 1–10° incidence angles by conjugately multiplying the master and slave SLC images. After removing the flat-earth effect according to the orbit vector and the reference surface, the interferometric phases are filtered via multi-looking to reduce the phase noise, which is conducted by just spatially averaging the conjugated image of two complex SLC images using an azimuth-range rectangular window [49]. The multi-looking numbers are 6 in azimuth and 1–6 in range corresponding to about 200 m × 200 m. We should point out that the flattened phase is not wrapped because the local SSH is unlikely to fluctuate more than some tens of meters and, thus, the interferometric phases would not exceed $2\pi$, so phase unwrapping is not needed. The filtered interferometric phase of each pixel can be transformed into the difference in ranges that the pixel refers to the two antennas, and, thus, two true ranges can be obtained, one of which is decided by the time delay.

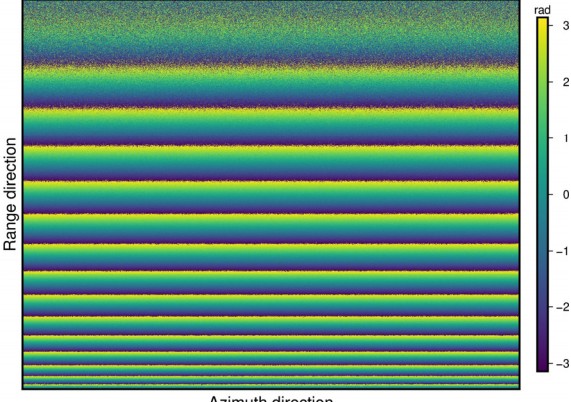

**Figure 2.** Example of TG2 InIRA interferometric phase image.

The geometric positioning of the TG2 InIRA image is implemented by solving the range-Doppler equation sets [47]:

$$|P_t - S_2| = r_2 = r_1 + \frac{\lambda}{2\pi}\phi \tag{1}$$

$$|P_t - S_1| = r_1 \tag{2}$$

$$-\frac{2}{\lambda r_1}[P_t - S_1] \cdot V_s = f_{dc} \tag{3}$$

where $P_t$ denotes the target, $S_1$ and $S_2$ denote the positions of two antennas (their connection forming the interferometric baseline B, whose inclination angle and length errors are estimated and corrected by using the MSS model), $r_1$ and $r_2$ are the distances between the target and the two antennas, $\phi$ is the interferometric phase (obtained by interferometric signal processing), $V_s$ is the speed of TG2 spacecraft, $f_{dc}$ is the Doppler centroid of azimuthal

signal after imaging processing, and λ is the wavelength of the carrier frequency. The 3-D coordinates of $P_t$ in the Earth-Centered Earth-Fixed (ECEF) coordinate system can be calculated, i.e., the geometric positioning of $P_t$, by solving Equations (1)–(3) either analytically or by the Iterative Newton method [47]. The above geometric positioning is conducted for every pixel of SAR image to obtain their 3-D coordinates in the ECEF coordinate system, which can be converted to the geodetic coordinates including longitude, latitude, and height relative to the WGS 84 ellipsoid (i.e., SSH).

The original grid resolutions of SSH obtained by TG2 InIRA are about 30 m along-track, and 30–200 m across-track. However, the spatial resolution is smoothed to uniform 200 m both along-track and across-track after multi-looked. The antenna pattern along with the 5° incidence angle as well as the quasi-specular scattering characteristics of the sea surface make the echo SNRs from the swath sides are lower than that from the swath center. Therefore, the echo data corresponding to 2.5–8° incidence angles are used, i.e., the used swath is about 35 km. Figure 3 presents an example of SSH image obtained in the Western Pacific along with the image showing the 2.5–8° incidence angles.

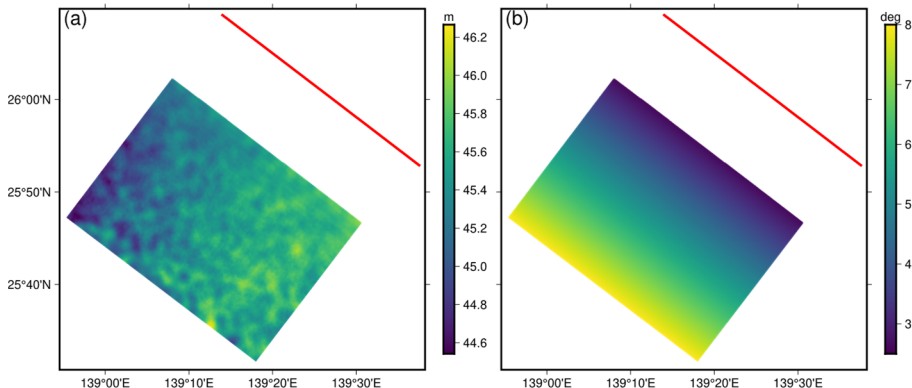

**Figure 3.** SSH image (**a**) and incidence angle image (**b**) of TG2 InIRA, the red line is the satellite nadir track. The incidence angles range from 2.5° to 8°.

### 2.3. Correction of Errors

The error sources of InIRA in SSH measurement mainly include ionosphere/troposphere delay errors, sea-state bias (SSB), geophysical errors, interferometric phase error, baseline error and system error [34,50–53]. The error sources of InIRA are quite different from those of CA although they have some common error sources, e.g., ionosphere/troposphere delay errors, geophysical errors and SSB. The influences of ionosphere/troposphere delay errors are different for InIRA and CA because on one hand the range beam of InIRA is quite larger than that of CA and thus both the ionosphere and troposphere may not be uniform, and on the other hand, this non-uniform may affect the baseline estimation. Geophysical errors mainly include those from tide modeling, inverse barometer effect, etc. In this work, we use the DOV to calculate the gravity anomaly based on the fact that the above long wavelength errors have very little influence on gravity recovery for this approach [22]. In addition, the simultaneous wide-swath measurement of the InIRA also guarantees the high relative accuracy of SSH measurement and high correlation of sea state, which is quite different from the independent waveform measurements of different CAs. And this characteristic also reduces the influence of the long wavelength errors.

As for the SSB, the sea-state directly affects the waveform of CA echo corresponding to the pulse-limited sea surface and, thus, the range measurement because accurate range measurement is realized according to the waveform. But for the InIRA, the sea-state mainly affects the distribution of interferometric phases of scattering centers within the spatial resolution cell and this effect can be greatly reduced via the Gaussian filtering because the echoes are corresponded to both beam-limited and range-gated sea surface. The Gaussian filtering is described in the next paragraph.

What error sources the InIRA have different from CA in SSH measurement are exactly the interferometric phase error and the baseline error. These two errors have the most notable influences on gravity recovery, so special algorithms should be designed for correcting these two kinds of errors. The interferometric phase error is caused by thermal decorrelation, geometric decorrelation, and angular decorrelation [34]. It can be smoothed by two-dimensional Gaussian filtering. Here, the filter window size is 65 along-track and 10–65 across-track corresponding to about 2 km × 2 km. The filter parameter is decided with $3\sigma$ corresponding to about 1 km. The Gaussian filtering is used for simultaneously suppressing the SSB effect and interferometric phase error.

The baseline error induced SSH measurement error has two contributions, i.e., the baseline roll error (BRE) and the baseline length error (BLE). According to the geometry of interferometric measurement as shown in Figure 4a, the height $h$ of point $P$ above a reference ellipsoid can be calculated by Equation (4) after the spacecraft orbit height $H$, the range $r_1$ and the look angle $\theta$ referring to $P$ have been made available:

$$h = H - r_1 \cos\theta \tag{4}$$

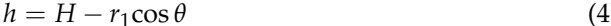
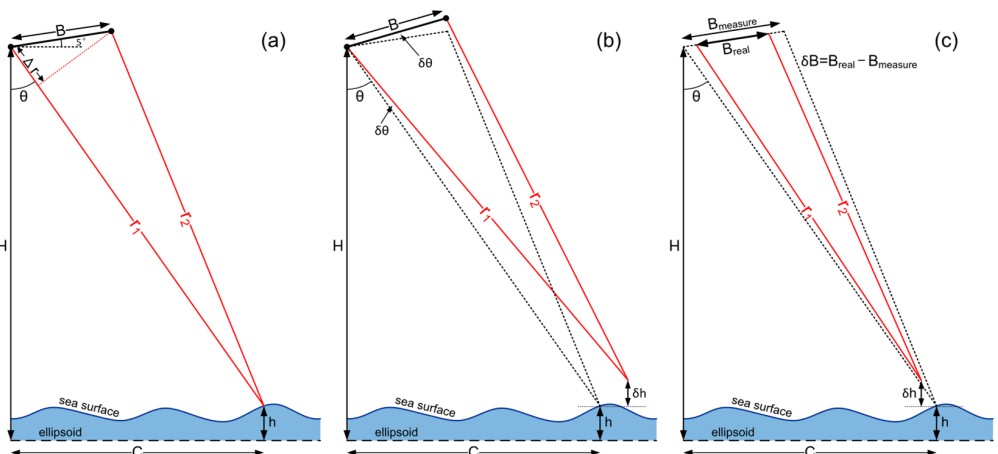

**Figure 4.** Diagram of (**a**) the interferometric measurement concept, (**b**) height error $\delta h$ caused by BRE $\delta\theta$, (**c**) height error $\delta h$ caused by BLE $\delta B$. $H$ is the ellipsoidal height of satellite orbit, $h$ is height of the sea surface point $P$, $C$ is the across-track distance from nadir to $P$, $\theta$ is the look angle, $B$ is the baseline length, $r_1$ and $r_2$ are the distances from two antennas to $P$, $\Delta r$ is the distance difference of $r_1$ and $r_2$.

As shown in Figure 4b, since the angle between the antenna beam pointing and the baseline is almost constant, the baseline roll knowledge error is just the look angle error. Thus, the roll knowledge error will introduce a height error $\delta h$ at $P$ within the swath as [34]

$$\delta h = r_1 \sin\theta \delta\theta \approx C\left(1 + \frac{H}{R_E}\right)\delta\theta \tag{5}$$

where $C$ is the across-track distance from the nadir point to $P$, $\delta\theta$ is the knowledge error of the baseline roll, and $(1 + H/R_E)$ is the correction term that considers the curvature of the earth, where $R_E$ is the radius of the Earth. As shown in Figure 4c, the baseline length knowledge error $\delta B$ will introduce a height error $\delta h$ as

$$\delta h = -\frac{r_1 \sin\theta \tan(\theta - 5°)}{B}\delta B \approx -\left(\frac{C^2}{H} - C\tan 5°\right)\left(1 + \frac{H}{R_E}\right)\frac{\delta B}{B} \tag{6}$$

where $B$ is the baseline length. Note that Equations (5) and (6) apply when the look angle and the baseline inclination are both small (e.g., the baseline inclination of TG2 InIRA is 5°). If the roll angle and baseline length can be perfectly measured, their induced SSH errors can all be perfectly corrected. Consequently, the realistic SSH errors brought by knowledge

errors of the roll angle and the baseline length are indeed the residual errors stemming from thermal deformation and mechanical resonance of the baseline. In fact, both the BRE and BLE are largely brought by platform, while the gyroscope measurement error can introduce roll error as well [54]. Equations (5) and (6) show that the BRE leads to a linear SSH error along the across-track direction, while the BLE leads to a quadratic SSH error along the across-track direction.

In the gravity recovery experiment, the correction method has been designed based on the empirical local estimation technique [54,55]. The baseline errors not only vary with time $t$ in the along-track direction, but also vary with the attitude in the across-track direction. In Equation (7), the SSH measurement $h_{obs}(C,t)$ is decomposed as the true SSH $h_{real}(C,t)$, the BRE contribution (linear term), the BLE contribution (quadratic term) and the sum of all other errors $\epsilon(C,t)$,

$$h_{obs}(C,t) = h_{real}(C,t) + \left(1 + \frac{H}{R_E}\right)\left(\delta\theta(t) + \frac{\delta B(t)}{B}\tan 5°\right) \cdot C - \left(1 + \frac{H}{R_E}\right)\frac{\delta B(t)}{HB} \cdot C^2 + \epsilon(C,t) \tag{7}$$

In practice, we first use a static reference $h_{ref}$ to replace the $h_{real}$, e.g., we use the Shuttle Radar Topography Mission (SRTM, [56]) derived digital elevation model over land and use the mean sea surface (MSS) model MSS_CNES_CLS2015 [57] over ocean. Then, according to the difference between $h_{obs}$ and $h_{ref}$, the BRE and BLE parameters of each pass are optimized and adjusted to correct the residual error [54]. The whole SSH data from all available orbits can be corrected with a unified standard by this approach.

In this work, the MSS model is only used to provide a stable reference surface for estimation and correction of baseline errors for high accuracy SSH reconstruction, there is no influence on the acquisition of instant SSH. The positions of the two antennas can be more accurate after correction of the baseline error, which is very crucial for accurately reconstructing the 3-D coordinates of the sea surface pixel via solving the range-Doppler equation sets. Although this approach may lead to partial loss of the ocean dynamic component, it is appropriate for gravity recovery because the dynamic component of the sea surface is indeed the interfering factor that should be removed. If the dynamic sea surface is the key information to obtain for oceanographic applications, e.g., eddy detection and tracking [58], the baseline correction should be much more carefully handled when using the MSS model [59], which is out of the scope of this work.

To evaluate the effectiveness of baseline correction and to show the effect of high-frequency oscillation of the baseline, the along-track and across-track residual geoid slopes are calculated after 2 km × 2 km SSH resampling (the resampling method will be introduced in Section 3.2). The statistical results of 71 passes are listed in Table 1. As it is shown, the STD of the along-track slope and the mean of the across-track slope have been remarkably reduced from 11.047 to 5.248 μrad and from −5.598 to 0.544 μrad, respectively. In addition, for better visually see the difference without and with baseline correction, the residual geoid height of a sample pass is presented in Figure 5; as can be clearly seen, the high-frequency oscillation has been deleted and the details are outstood.

**Table 1.** The improvement of baseline correction on residual geoid slope.

| | Unit: μrad | Max | Min | Mean | STD | RMSE |
|---|---|---|---|---|---|---|
| without baseline correction | along-track slope | 78.427 | −89.620 | −0.049 | 11.047 | 11.047 |
| | across-track slope | 49.282 | −57.396 | −5.598 | 8.406 | 10.100 |
| after baseline correction | along-track slope | 22.715 | −24.923 | 0.002 | 5.248 | 5.248 |
| | across-track slope | 42.463 | −37.700 | 0.544 | 7.181 | 7.201 |

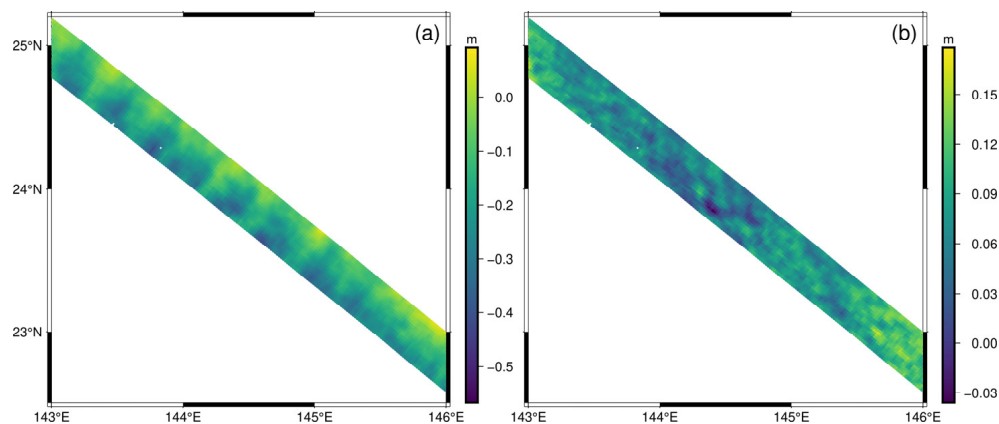

**Figure 5.** Residual geoid height of a TG2 InIRA sample pass, (**a**) without baseline correction, (**b**) after baseline correction.

We should emphasize that the above BRE and BLE correction is specifically designed for TG2 InIRA in consideration of the following three facts: (1) the baseline is 5° inclined, (2) single swath, (3) the phase centers of the two antennas are almost at the geometrical centers of the two slotted waveguide array antennas, which forms a "direct interferometric baseline", so the interferometric paths are quite stable. As for the SWOT, its baseline is horizontal, two swaths are observed and reflectarray antennas are adopted with feeds installed on the satellite, forming an "indirect interferometric baseline", so the interferometric paths are relatively longer and may be affected by satellite itself, i.e., the BRE and BLE corrections can be a little bit complex. However, the double-swath brings about the good opportunity for using the symmetric characteristics to correct the BRE and BLE.

## 3. Experimental Area and Data Allocation

### 3.1. Experimental Area

The orbit of Tiangong-2 space laboratory is not designed for geodetic mission of gravity recovery on purpose, and the TG2 InIRA did not work at full orbit time [39]. Although the observations mainly cover the South China Sea and the Western Pacific region, some observation gaps still exist. Considering the experimental area should have an approximately equivalent amount of ascending and descending passes, and at the same time have typical submarine structures, an experimental area (139°E–146°E, 21°N–28°N) in the Western Pacific is selected, which is located at the boundary between the Pacific and Philippine plates, and has submarine structures such as seamounts, islands and trenches. The biggest trench has a long north–south trend making the east component of DOV sensitive to the SSH variation, which is very appropriate for subsequent verification and comparison of the results. In addition, due to the geographical significance of this area, many shipborne gravity measurements have been carried out since 1960s, which provide us the necessary high-quality validation data for gravity recovery experiment. Figure 6a shows the 71 nadir tracks of TG2 InIRA over the experimental area during the period from December 2016 to February 2019, including 42 ascending passes and 29 descending passes. Figure 6b shows the number of repeats of TG2 InIRA swaths. And Figure 6c is the bathymetry map of the study region for reference.

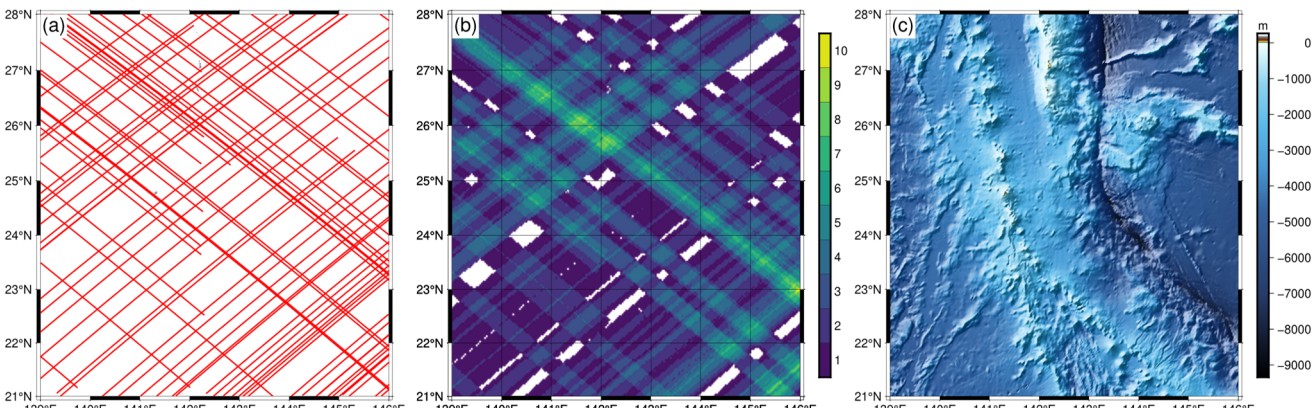

**Figure 6.** Schematic of 71 passes of TG2 InIRA over the experimental area, (**a**) the nadir tracks, (**b**) the number of repeats. And (**c**) is the bathymetry of the study region.

### 3.2. SSH Resampling

The SSH data of TG2 InIRA need to be preprocessed before used for gravity recovery. First, gross errors need to be removed. The observation data may be polluted by rainfalls or other unexpected events; therefore, the measured interferometric phases can be degraded. Thus, the resulted SSH should not be used. Therefore, any SSH anomaly greater than 5 m is considered as an outlier and removed. Here, we use the MSS_CNES_CLS2015 model as the reference datum and the outlier exclusion rate for the 71 passes data is about 0.06%. We should point out that this threshold is not very critical, e.g., if 2 m or 3 m is chosen as the threshold, it does not make an obvious difference to the final results.

Then the SSH data of all passes are resampled into 2 km × 2 km grids to reduce the data volume. In the implement of resampling, we first use the ephemeris data of the TG2 spacecraft orbit to generate the coordinates of the nadir points. In the along-track direction, the nadir points are resampled according to 2 km interval, and in the across-track direction, the sampling points are also taken at the same 2 km interval. Thus, the 2 km × 2 km resampling grids can be generated in this way, as shown in Figure 7a. The resampled SSH data of TG2 InIRA are then generated according to the resampling grids, as shown in Figure 7b. To reduce the influence of noise, all the SSH data of original grid resolutions falling into each 2 km × 2 km grid are averaged to obtain the final SSH for that grid. Of course, the number of the SSH data of original grid resolutions falling in the near grids is different from that falling in the far grids; however, the difference can be ignored. Although the resampling process sacrifices the spatial resolution somehow, it effectively improves the accuracy of SSH.

A Gaussian low-pass filter is generally used by CAs to smooth SSH data so as to suppress the noise and improve the SNR [60]. Here, the same kind of low-pass filter is also used to filter the 2 km × 2 km SSH data of TG2 InIRA after extending it from one-dimension to two-dimension. The adopted Gaussian low-pass filter has a gain of 0.5 at 6.7 km, and the window size is 15 × 15 (corresponding to about 30 km × 30 km), which are selected by referring to the works [16,60]. In addition, different from the one-dimensional filtering on CA SSH data, the filter operation on WSA data has boundary effects on both sides of swath, i.e., the performance is usually degraded because the involved data are reduced. In the field of image processing, although there are some basic methods available for alleviating this effect, such as zero padding and symmetric padding, here, we simply truncate the filter at the swath boundary.

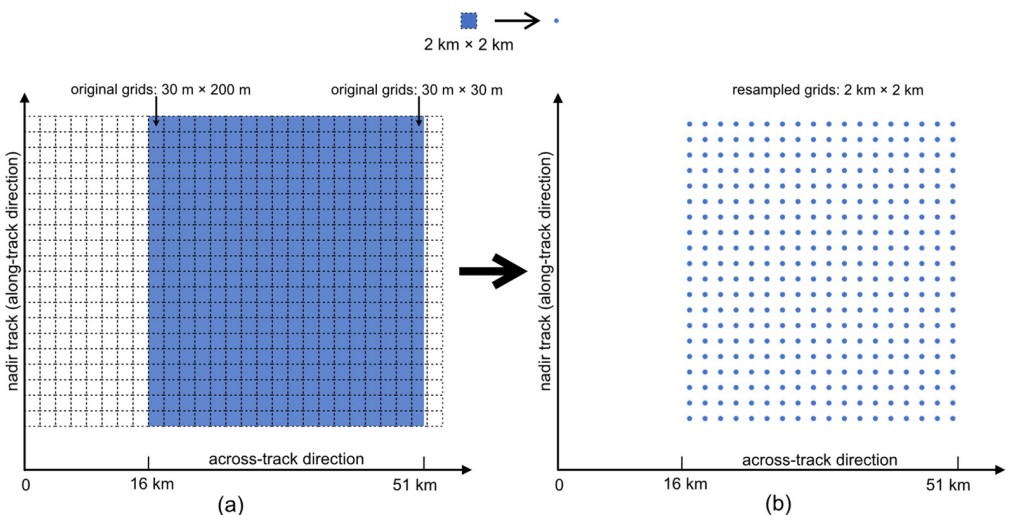

**Figure 7.** Resampling of TG2 InIRA SSH data at 2 km × 2 km grids. (**a**) Original grid resolution SSH (each blue square contains dense observation points). (**b**) Resampled SSH at 2 km × 2 km grids.

*3.3. Data Allocation*

A total of 326,841 observation data are obtained after all the 71 TG2 InIRA passes' data are resampled into 2 km × 2 km grids. For better comparing the performances of TG2 InIRA data and that of CA, four CAs of different inclinations are taken, i.e., Jason-1 (J1), Jason-2 (J2), SARAL (SA) and CryoSat-2 (C2). The geophysical data record (GDR) data of J1, J2 (20 Hz) and SA (40 Hz) are downloaded from the AVISO, and the 20 Hz GDR data of C2 are downloaded from the ESA. The 20 Hz and 40 Hz altimetry data are down sampled to 4 Hz (i.e., the along-track interval is about 2 km) for matching to the 2 km × 2 km grids of TG2 InIRA. In addition, the 1 Hz error corrections are interpolated to 4 Hz correspondingly and applied, such as the ionosphere, dry and wet troposphere, SSB and tide corrections.

The data information of the above altimeters is provided in Table 2, as it is shown 111,409 of J1 and 186,896 of J2 data are allocated from their geodetic missions, and 233,557 of SA data and 233,419 of C2 data are allocated as well. We should illustrate that these data are allocated by considering both the spatial coverage and approximately equal amount of different CAs. So, a total of 765,281 data from four altimeters' 2502 passes are obtained. It should be noted that the total TG2 InIRA data are 32,6841 covering about 85.5% of the study area, i.e., there are some gaps mainly due to the orbital coverage. It is reasonable to have the same gaps for CAs, so there are about 654,315 (765,281 × 85.5%) data fall into the observed area by TG2 InIRA, which is almost two times of TG2 InIRA data. The temporal variation of gravity field is generally considered to be very slow and small, so we do not consider the time datum of different altimeter data in the experiment.

**Table 2.** Summary of altimeter data used in this study.

| Altimeter | Duration | Inclination | Passes | Valid Points |
|---|---|---|---|---|
| J1 | May 2012–June 2013 | 66° | 363 | 111,409 |
| J2 | July 2017–October 2019 | 66° | 608 | 186,896 |
| SA | July 2016–January 2019 | 98.5° | 823 | 233,557 |
| C2 | July 2010–January 2013 | 92° | 708 | 233,419 |
| J1J2+SA+C2 | / | / | 2502 | 765,281 |
| TG2 | December 2016–February 2019 | 43° | 71 | 326,841 |

If we simply compare the 71 passes of TG2 InIRA to the 2502 passes of CAs, the high observation efficiency of WSA is very clear. The measured SSHs in the study region by TG2 InIRA, J1J2, SA and C2, are presented in Figure 8a,c–e, respectively, and Figure 8b

presents the fused SSH of the four CAs' observations. As can be seen from Figure 8c–e, the features of different orbital inclinations are clearly exhibited. The fused SSH of Figure 8b looks much more uniformly distributed than the SSH of TG2 InIRA as shown in Figure 8a.

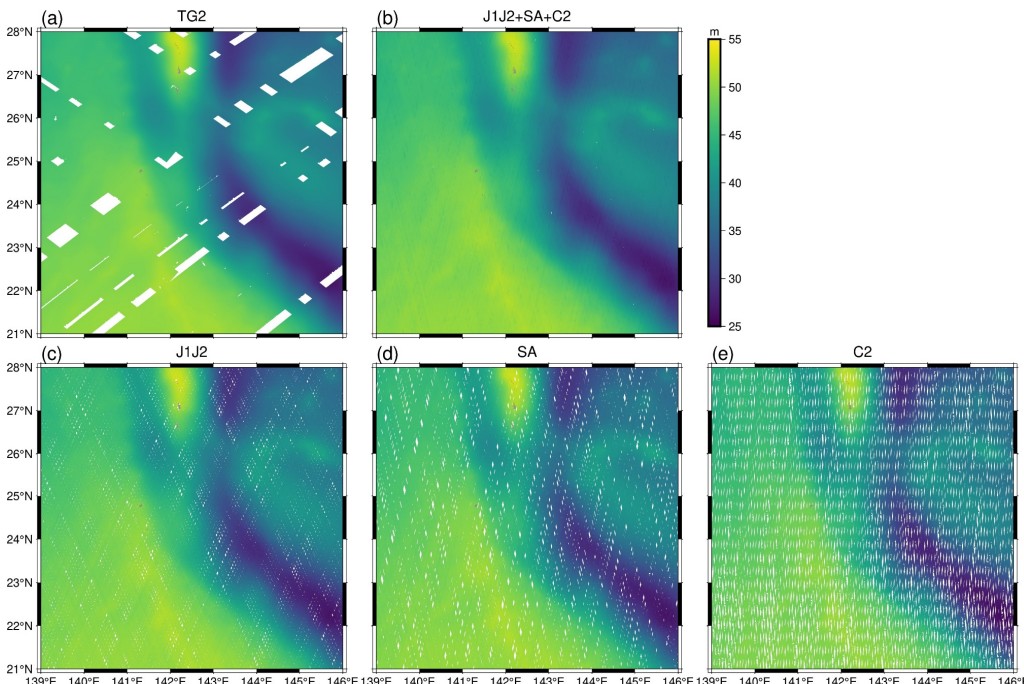

**Figure 8.** Measured SSHs by TG2 InIRA and CAs. (**a**) TG2, (**b**) J1J2+SA+C2, (**c**) J1J2, (**d**) SA, (**e**) C2.

## 4. Methodology

### 4.1. Determination of DOV

As shown in Figure 1, the SSH measured by satellite altimeters can be expressed as the sum of geoid height *N*, dynamic ocean topography (DOT) *ζ* and error *e* [61],

$$SSH = N + \zeta + e \qquad (8)$$

where the geoid height *N* can be further represented by a reference geoid height model and residual geoid height. Similarly, the DOT can be described in terms of a mean dynamic topography (MDT) and a time varying ocean topography. In oceanography, the DOT is the important signal, while in geodesy the geoid height (or the geoid slope) is of prime interest.

In this study, we use the EGM2008 [62] as the reference geoid model, and the DTU15MDT [63] as the MDT model. As the widely recognized high-precision Earth gravitational field model, EGM2008 has been selected as the reference model in many studies, such as [38,64,65]. And the DTU15MDT, as a high-precision global MDT model, has also been selected as the MDT model in many studies, such as [15,66]. The resolutions of EGM2008 and DTU15MDT are about 5′ and 7.5′, respectively. We need to interpolate the EGM2008 and the DTU15MDT into the 1′ × 1′ grids, which are then linearly interpolated to 2 km × 2 km grids of TG2 InIRA. Indeed, the 1′ × 1′ grids of EGM2008 can be conveniently obtained by calculation of gravity field functionals based on ellipsoidal grids which is provided by International Centre for Global Earth Models (ICGEM) [20,67]. And the 1′ × 1′ grids of DTU15MDT can be downloaded from the Technical University of Denmark (DTU) site directly.

There are usually two approaches to derive the marine gravity anomalies, one is by geoid height directly and the other is by DOV [16]. Geoid height is the distance of the geoid relative to the reference ellipsoid, which can be converted to marine gravity anomalies by inverse Stokes equation [68]. DOV is the angle between the plumb line of the geoid and the normal line of the reference ellipsoid, which can be further decomposed into two mutually perpendicular components: the north–south (*ξ*) and the east–west (*η*), and based on these

two components, the gravity anomalies can be derived via the inverse Vening Meinesz formula [18] or the Laplace equation [10]. The approach via the geoid height requires absolute SSH measurement, while the approach via DOV by the inverse Vening Meinesz formula and Laplace equation only needs relative SSH measurement [22]. The relative measurement not only reduces the influence of orbit determination error, but also reduces the influence of the troposphere and ionosphere delay error [16]. In addition, the phase measurement characteristic of the WSA also ensures a high relative SSH measurement accuracy. Therefore, the DOV approach is adopted in this study.

The remove–restore method is usually adopted to calculate the DOV and gravity anomalies, which performs well on reducing the effect of long wavelength error [69,70]. In this method, the residual geoid height is obtained by removing the EGM2008 geoid height model and based on which, the residual DOV and residual gravity anomalies can be calculated. Finally, the modelled DOV and gravity anomalies of EGM2008 are added back to obtain the complete DOV and gravity anomalies.

Residual DOVs can be calculated according to the residual geoid slope. For CAs, the along-track residual DOV $\varepsilon$ is calculated by differentiating two adjoining measured residual geoid heights $N_{res}$ along altimeter track [70],

$$\varepsilon = \frac{\partial N_{res}}{\partial s} \tag{9}$$

where $\partial N_{res}$ is the variation of residual geoid height over $\partial s$ which is generally the resampling interval along the track. The along-track residual DOV can be further decomposed into the north and east components by

$$\varepsilon_i + v_i = \xi_q \cos \alpha_i + \eta_q \sin \alpha_i, i = 1, 2, \cdots, M \tag{10}$$

where $\xi_q$ and $\eta_q$ are, respectively, the north and east components of residual DOVs at regular grid point $q$, $v_i$ is the residual error, $\alpha_i$ is the azimuth angle of $\varepsilon_i$, $M$ is the number of along-track residual DOVs. For WSA data, Equations (9) and (10) are applicable both in the along-track direction and across-track direction. Figure 9 presents the calculation of the azimuth angle for CA and TG2 InIRA. Hwang et al. [70] introduced a weighted least squares (WSL) method to determine the components $\xi$ and $\eta$ which is also adopted in this study. And the weights $P_i$ for the observations are set simply according to the distance, i.e., $P_i = 1/d_i$, where $d_i$ is the distance from the observation point to the grid point. To reduce the influence of noise, the gross errors of $\varepsilon_i$ by different altimeters are separately excluded using the $3\sigma$ rules. In addition, benefiting from the flexibility of Equations (9) and (10), the TG2 InIRA data can be directly merged with the CAs data in order to evaluate the fused gravity recovery of WSA and CA.

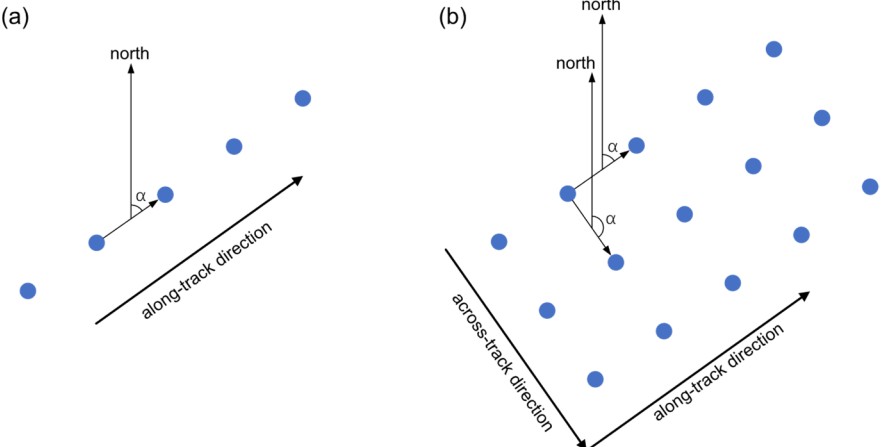

**Figure 9.** Calculation of azimuth angle $\alpha$. (**a**) CA. (**b**) TG2 InIRA.

There is a common problem for the past and the current CAs aiming for marine gravity recovery: the accuracies of the north and east components of DOVs are unbalanced and vary with latitude. And since the satellite orbit inclinations are quite large (J1J2: 66°, SA: 98.5°, C2: 92°), the uncertainty of the east component is several times greater than that of the north component at low latitudes [13]. However, there is no such problem for TG2 InIRA because both the along-track slope and across-track slope can be calculated independent of the latitude and inclination of the orbit. This characteristic of TG2 InIRA, on one hand, helps to improve the accuracy of the east components and on the other hand, helps to obtain consistent accuracies of the north and east components, which is beneficial to gravity recovery.

In this paper, our purpose is to compare the gravity recovery performance of TG2 InIRA and CAs, so we only need to ensure the same calculation method is used. The weighted least squares method not only has high gravity recovery accuracy, but also has higher computational efficiency as well, which is good for processing large amount of data. So it is adopted in this study although there are some more accurate methods for the DOV calculation [22,70,71] by which the recover accuracy of gravity anomalies can be further improved, the complexity and the computational cost are both much higher.

### 4.2. Gravity Anomalies Recovery

In the following, we use the inverse Vening Meinesz formula [18] to calculate the residual gravity anomalies from the residual DOVs.

$$\Delta g_{res}(p) = \frac{\gamma_0}{4\pi} \iint_\sigma H'\left(\psi_{qp}\right) \left(\xi_q \cos \alpha_{qp} + \eta_q \sin \alpha_{qp}\right) d\sigma_q \tag{11}$$

where $p$ and $q$ are two points separated by a spherical distance of $\psi_{qp}$ on the unit sphere. $\Delta g_{res}(p)$ is the residual gravity anomaly at $p$, $\gamma_0$ is the normal gravity, $\xi_q$ and $\eta_q$ are, respectively, the north and east components of the residual DOV at $q$, $\alpha_{qp}$ is the azimuth angle from $q$ to $p$. $H'\left(\psi_{qp}\right)$ is the kernel function of spherical distance between $q$ and $p$ defined as,

$$H'\left(\psi_{qp}\right) = -\frac{\cos \frac{\psi_{qp}}{2}}{2\sin^2 \frac{\psi_{qp}}{2}} + \frac{\cos \frac{\psi_{qp}}{2} \cdot \left(3 + 2\sin \frac{\psi_{qp}}{2}\right)}{2\sin \frac{\psi_{qp}}{2} \cdot \left(1 + \sin \frac{\psi_{qp}}{2}\right)} \tag{12}$$

Due to the singularity of the kernel function at zero spherical distance, the innermost zone contribution $\Delta g_{inn}(p)$ around the neighborhood of the point $p$ must be included in gravity anomaly calculation and it can be computed by

$$\Delta g_{inn}(p) = \frac{S_0 \gamma_0}{2} \left(\xi_y + \eta_x\right) \tag{13}$$

where $\xi_y$ and $\eta_x$ can be respectively obtained by numerically differentiating $\xi$ and $\eta$ at point $p$ along the $y$ and $x$ directions. $S_0$ is the radius of the innermost zone given by

$$S_0 = \sqrt{\frac{\Delta x \Delta y}{\pi}} \tag{14}$$

where $\Delta x$ and $\Delta y$ are the grid sizes of residual DOV along the east and north directions, respectively. Having the north and east components of DOV been obtained at regular grids, Equation (11) can be implemented via fast Fourier transform (FFT). After restoring the EGM2008 gravity model $\Delta g_{EGM2008}$ back to the residual gravity anomalies, the complete marine gravity anomalies can be calculated by

$$\Delta g = \Delta g_{EGM2008} + \Delta g_{res} + \Delta g_{inn} \tag{15}$$

The data processing procedure for gravity recovery using the derived SSH from TG2 InIRA and that from CA is shown in Figure 10.

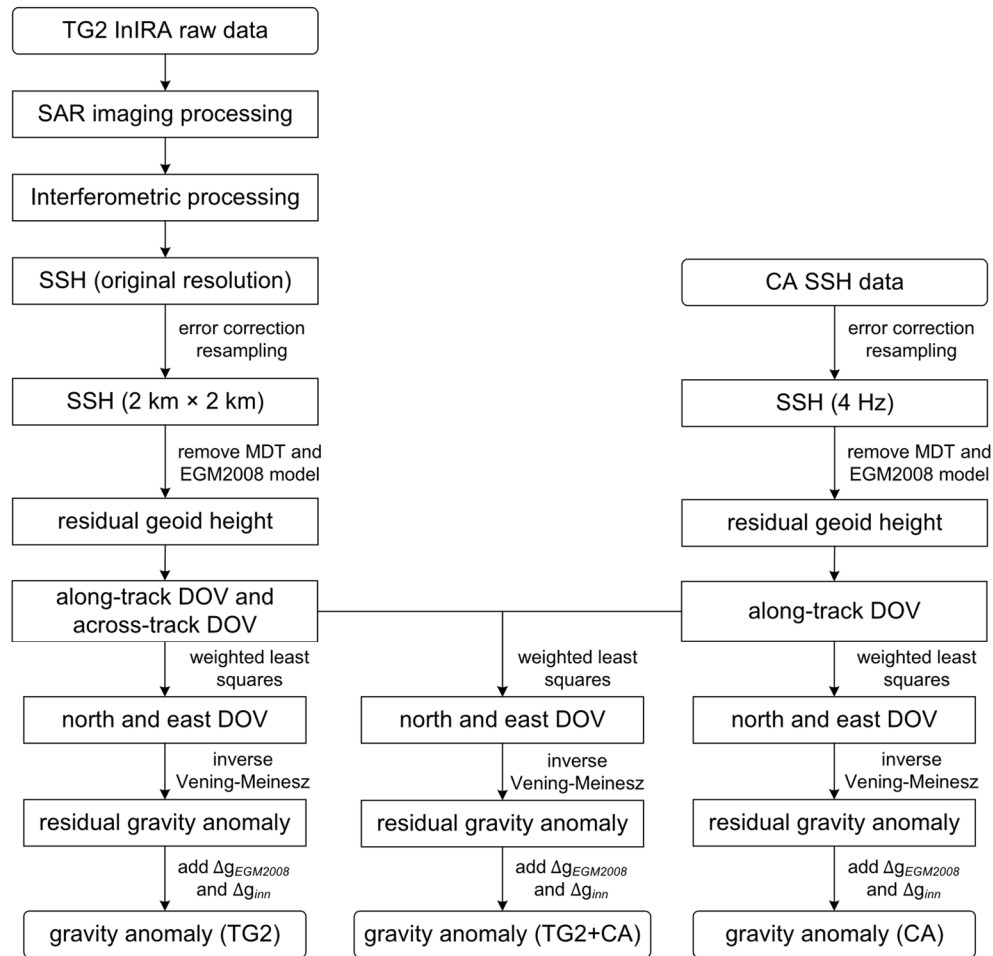

**Figure 10.** Flowchart of gravity recovery procedures using the derived SSH from TG2 InIRA and that from CA.

In order to guarantee the operation of FFT, the gapped areas of TG2 InIRA DOV are first filled by using the interpolation function *surface* of the Generic Mapping Tools (GMT, [72]), which uses continuous curvature splines in tension [73]. And then the gapped areas are removed after the gravity anomalies are calculated. In the next section, we present both the calculated DOVs as well as the gravity anomalies on $1' \times 1'$ geographic grids from TG2 InIRA and compare them with results of J1J2, SA and C2. We should illustrate that the gapped observation areas of TG2 InIRA have also been kept gapped for CA results, published gravity models and shipborne gravity data.

## 5. Results and Discussion

### 5.1. Comparison of DOV

Figures 11 and 12 present, respectively, the north and east components of DOVs derived from TG2 InIRA, J1J2, SA, C2, the fusion of CAs (denoted as J1J2+SA+C2) and the fusion of TG2 InIRA and CAs (denoted as TG2+J1J2+SA+C2). As shown in Figures 11 and 12, the north components and the east components of DOVs by TG2 InIRA and other CAs are visually consistent with each other, and it is true for the two fused results. If we compare the images of Figures 11 and 12 carefully, we can see the TG2 InIRA results exhibit higher spatial resolution than the CAs results do, especially for the north component. Meanwhile, the east component by C2 exhibits distinct north–south streaks due to the large orbital inclination.

Figures 13 and 14 present the different residual DOVs, showing that the results of TG2 InIRA differ significantly from that of CAs with finer textures exhibited resulting from higher spatial resolutions for both the north and east components. As for CAs, the

north components of J1J2, SA, C2 and their fusion are consistent, but bigger noise appears in the east component image due to the large orbital inclination, especially for C2. As for the results of TG2+J1J2+SA+C2, the north component is more consistent with that of J1J2+SA+C2, while the east component is more consistent with that of TG2 InIRA.

CAs are of relatively lower spatial resolution due to the larger footprint, but of high accuracy and stable measurement along track. TG2 InIRA lacks sufficient spatial coverage and repeated observations to reach the swath averaged accuracy [34] because it had been in-orbit for just about 27 Months. But its spatial resolution is higher (it mainly depends on the raw resolution of radar image and the available multi-looking numbers for suppressing the random noise, and the realization of which critically depends on the uniform coverage of repeated observations). The two altimeters have their own advantages and features, so their performances on DOV and gravity anomalies can be different. For example, the gravity information of the oceanic ridge patterns in the experimental area has been clearly outlined by CAs, but not by TG2 InIRA due to insufficient repeated observations as shown in Figure 6b.

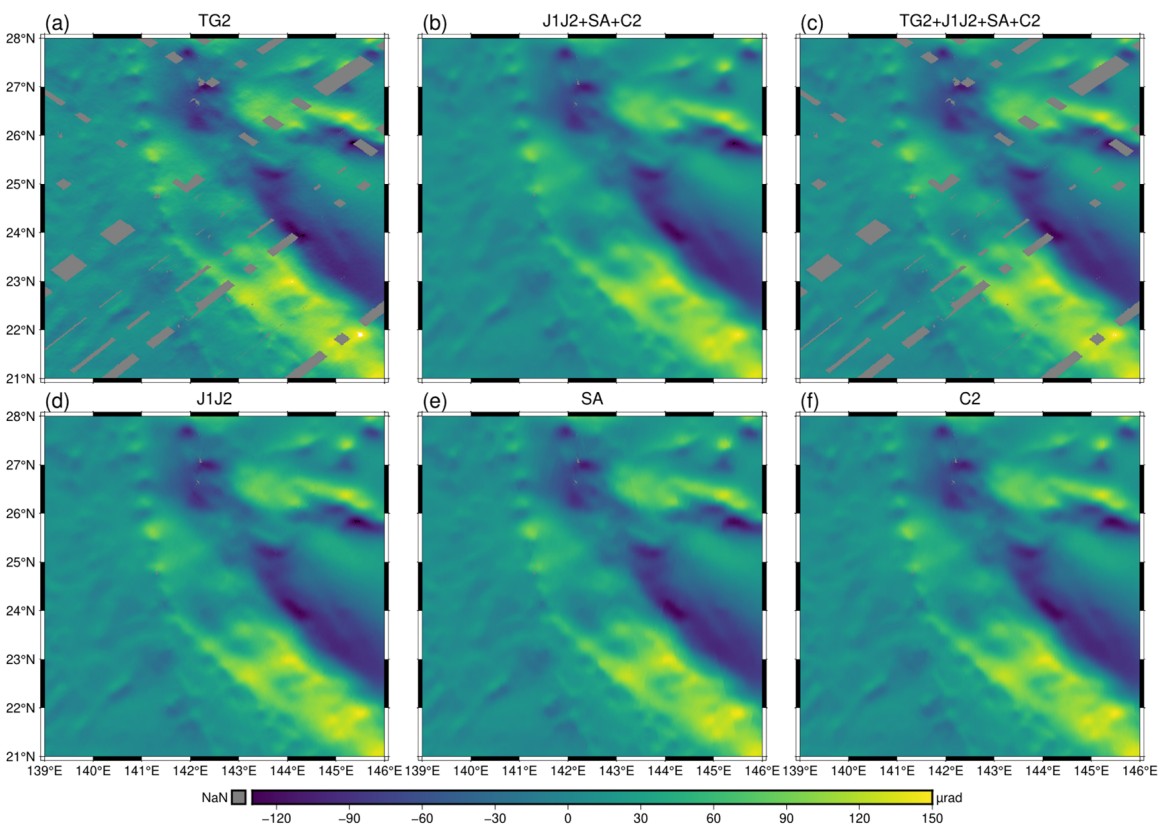

**Figure 11.** North component of the DOV obtained by TG2 InIRA and CAs, (**a**) TG2, (**b**) J1J2+SA+C2, (**c**) TG2+J1J2+SA+C2, (**d**) J1J2, (**e**) SA, (**f**) C2.

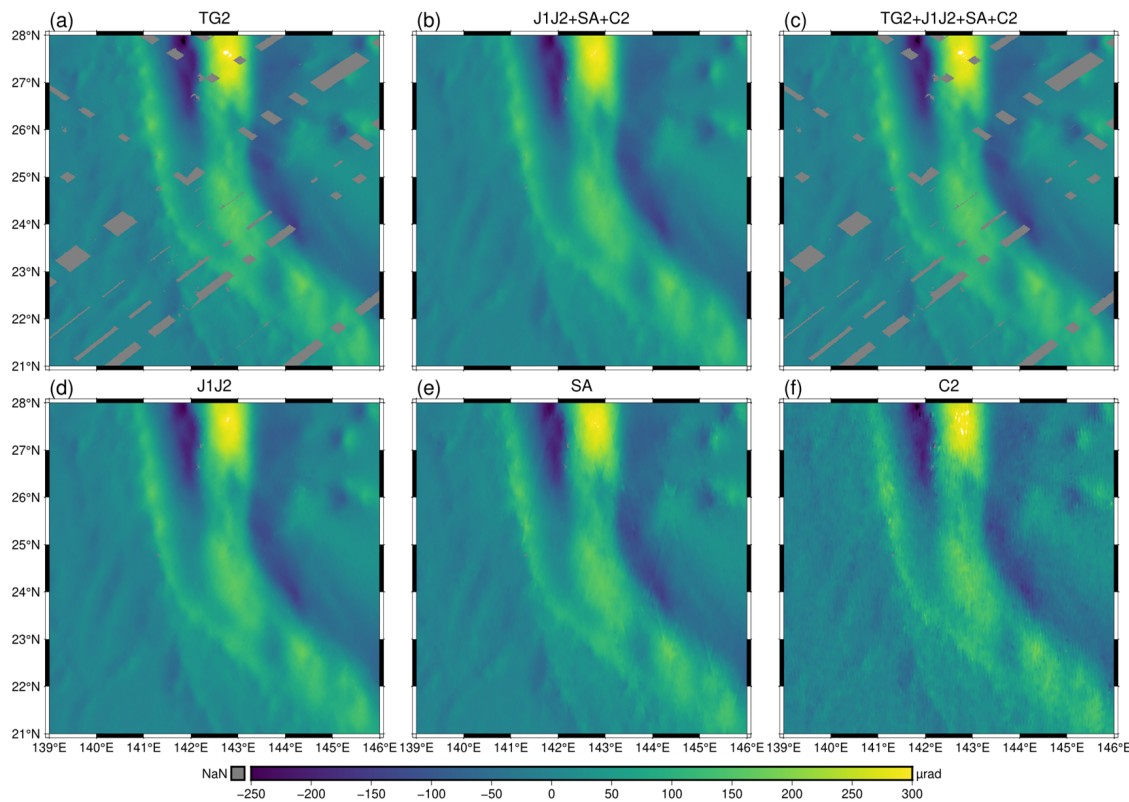

**Figure 12.** East component of the DOV obtained by TG2 InIRA and CAs, (**a**) TG2, (**b**) J1J2+SA+C2, (**c**) TG2+J1J2+SA+C2, (**d**) J1J2, (**e**) SA, (**f**) C2.

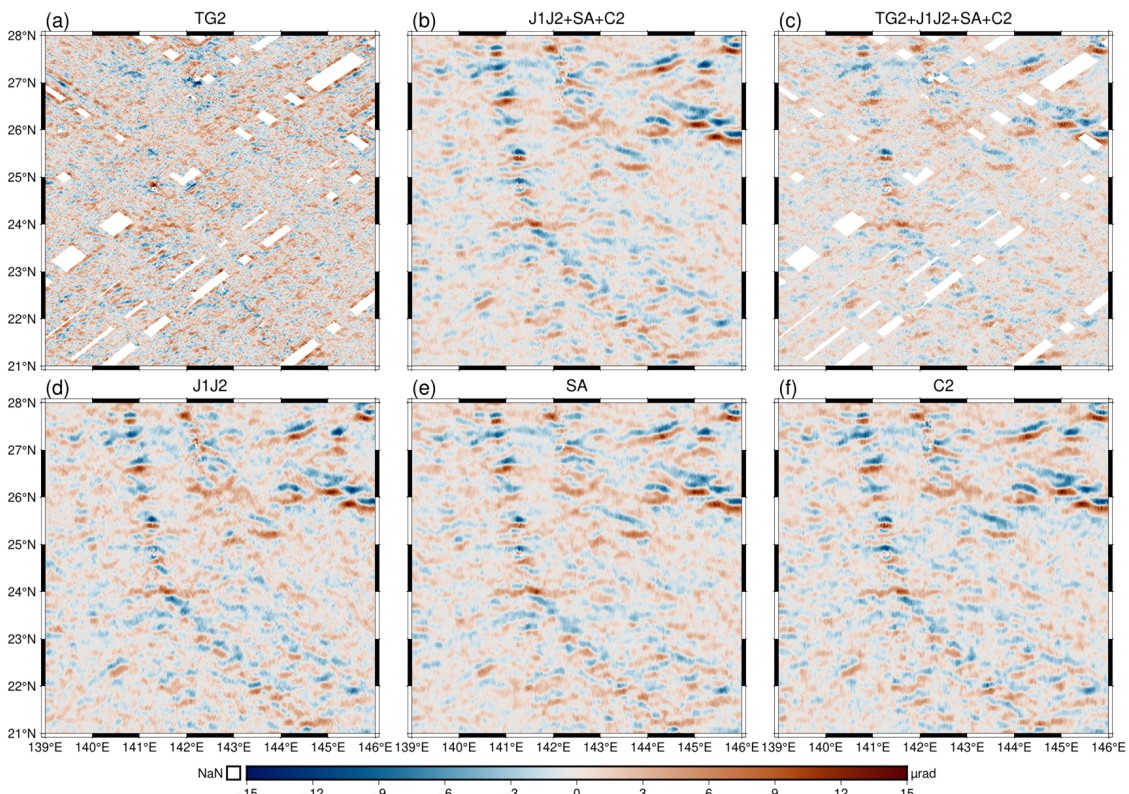

**Figure 13.** North component of the residual DOV obtained by TG2 InIRA and CAs, (**a**) TG2, (**b**) J1J2+SA+C2, (**c**) TG2+J1J2+SA+C2, (**d**) J1J2, (**e**) SA, (**f**) C2.

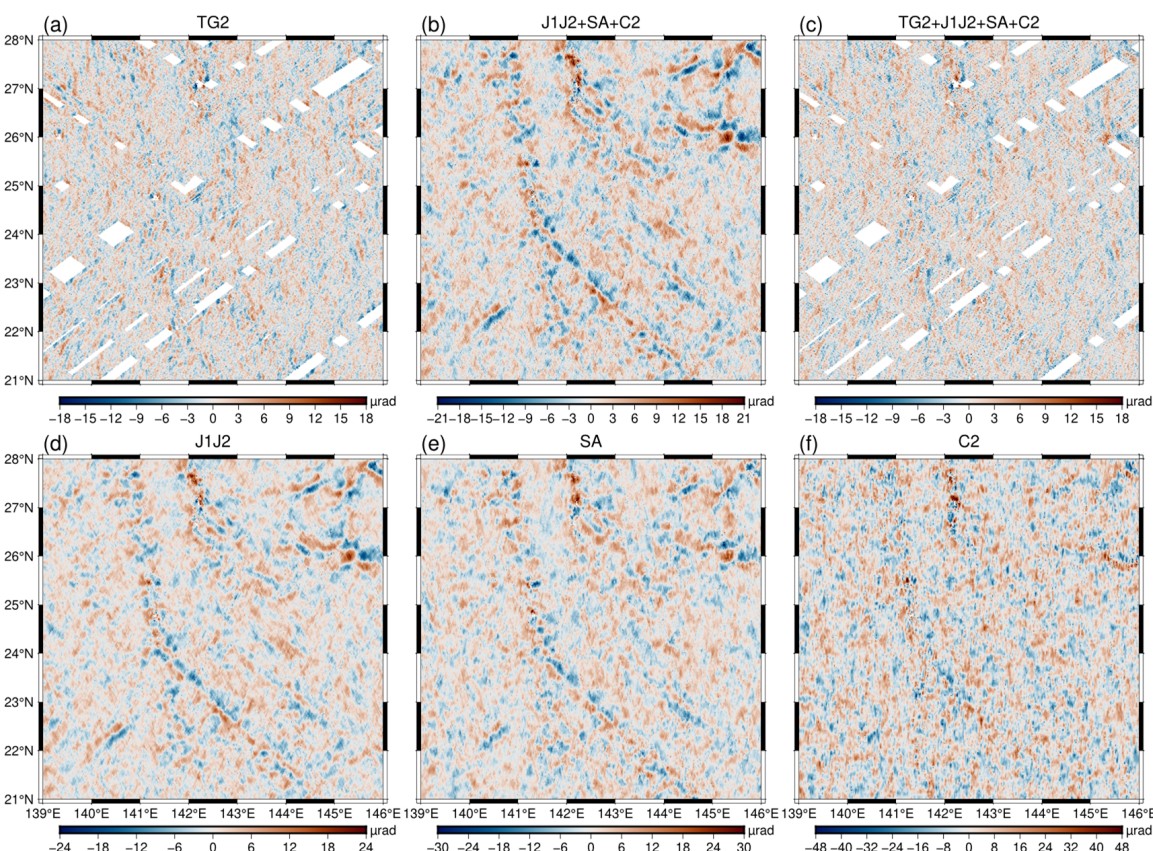

**Figure 14.** East component of the residual DOV obtained by TG2 InIRA and CAs, (**a**) TG2, (**b**) J1J2+SA+C2, (**c**) TG2+J1J2+SA+C2, (**d**) J1J2, (**e**) SA, (**f**) C2.

The statistics for the above Figures 13 and 14 are listed in Table 3 for clear and easy comparison. To compare the DOVs by TG2 InIRA with that by CAs more widely, the DOV model SIO V32.1 released by the Scripps Institution of Oceanography (SIO) is used for reference standard, and the detailed results are presented in Table 4. As shown in Table 4, for J1J2, SA and C2 altimeters, the STDs and RMSEs of the north components of DOVs are relatively consistent with each other, which are about 2.2 μrad, while the STDs and RMSEs of the east components gradually increase from 3.7 to 4.6 to 8.5 μrad as the orbital inclination varies from 66° to 98.5° to 92°. The STD and RMSE of the north DOVs obtained from J1J2+SA+C2 are close to those of the single-altimeter dataset, while the STD and RMSE of the east DOVs are about 3.8 μrad, which is greater than that of J1J2 but smaller than that of SA and C2, suggesting larger inclination altimeters degrade the quality of smaller inclination altimeters.

**Table 3.** Statistics of residual DOV for Figures 13 and 14.

| Dataset | North Component (μrad) | | | | | East Component (μrad) | | | | |
|---|---|---|---|---|---|---|---|---|---|---|
| | Max | Min | Mean | STD | RMSE | Max | Min | Mean | STD | RMSE |
| J1J2 | 15.976 | −15.016 | 0.059 | 2.057 | 2.058 | 26.667 | −26.982 | 0.000 | 3.455 | 3.455 |
| SA | 16.663 | −14.203 | 0.089 | 1.976 | 1.978 | 41.099 | −29.004 | −0.027 | 4.415 | 4.416 |
| C2 | 14.886 | −15.148 | −0.053 | 2.055 | 2.056 | 103.519 | −71.600 | −0.020 | 8.421 | 8.421 |
| J1J2+SA+C2 | 15.875 | −14.195 | 0.027 | 2.029 | 2.029 | 28.946 | −34.298 | −0.013 | 3.617 | 3.617 |
| TG2 | 21.689 | −19.075 | 0.510 | 2.793 | 2.839 | 20.527 | −17.219 | −0.135 | 2.756 | 2.759 |
| TG2+J1J2+SA+C2 | 16.493 | −14.520 | 0.161 | 1.906 | 1.912 | 24.348 | −23.469 | −0.105 | 2.839 | 2.841 |

**Table 4.** Comparison of DOV with the SIO V32.1 model.

| Dataset | Inclination | Valid Numbers | North Component (μrad) | | | | | East Component (μrad) | | | | |
|---|---|---|---|---|---|---|---|---|---|---|---|---|
| | | | Max | Min | Mean | STD | RMSE | Max | Min | Mean | STD | RMSE |
| J1J2 | 66° | 298,305 | 28.694 | −30.041 | 0.013 | 2.284 | 2.284 | 47.266 | −27.159 | −0.030 | 3.723 | 3.723 |
| SA | 98.5° | 233,557 | 33.839 | −28.383 | 0.043 | 2.227 | 2.228 | 59.412 | −38.139 | −0.058 | 4.585 | 4.586 |
| C2 | 92° | 233,419 | 27.271 | −29.203 | −0.099 | 2.294 | 2.296 | 104.765 | −80.368 | −0.050 | 8.522 | 8.522 |
| J1J2+SA+C2 | / | 765,281 | 28.332 | −29.404 | −0.018 | 2.236 | 2.236 | 49.632 | −37.477 | −0.044 | 3.842 | 3.843 |
| TG2 | 43° | 326,841 | 22.094 | −20.974 | 0.464 | 2.644 | 2.684 | 35.063 | −24.985 | −0.165 | 2.771 | 2.776 |
| TG2+J1J2+SA+C2 | / | 1,092,122 | 24.347 | −24.070 | 0.115 | 2.007 | 2.011 | 36.827 | −36.409 | −0.135 | 2.895 | 2.898 |
| EGM2008 | / | / | 27.122 | −25.454 | −0.046 | 1.532 | 1.533 | 32.797 | −19.564 | −0.030 | 2.010 | 2.010 |

As for the results of TG2 InIRA, although the STD of the north component is about 2.684 μrad, slightly larger than the 2.236 μrad of CAs, it is consistent to 2.776 μrad of the east component. As it is clearly shown, the STD of the east components of TG2 InIRA is obviously smaller than that of CAs, which is benefited mainly by the wide-swath capability and additional across-track measurement. We should point out that, as can be noticed, the mean deviations of the north and east components are about 0.464 μrad and -0.165 μrad, respectively. They are somewhat larger than that of CAs especially the north component. The main reasons can be explained as follows. The residual errors left by baseline correction will affect the mean deviation of across-track slope and, thus, the DOV, while the decomposition of the north and east components of DOV is related to the azimuth angle of satellite orbit, so these two components are affected differently by the residual errors for different orbits. As shown in Figure 15a,b, when the azimuth angle is in the orange area ($\alpha_s$: 0–45°, 135–225°, 315–360°), the mean deviation of the decomposed east component is larger than that of the north component. When the azimuth angle is in the blue area ($\alpha_s$: 45–135°, 225–315°), the case is reversed. For TG2 InIRA, the orbital inclination is 43°, the azimuth angle of TG2 InIRA in our experimental area is always in the blue area ($\alpha_s$: 45–135°), so the mean deviation of the north component is larger than that of the east component.

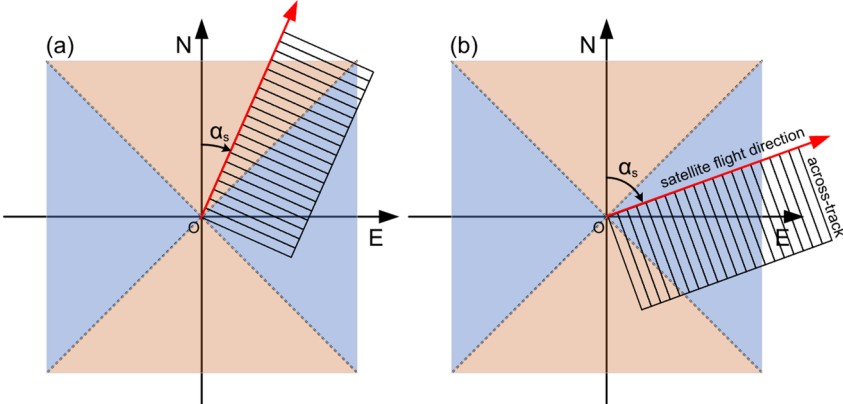

**Figure 15.** North and east components of DOV are affected differently by the azimuth angle $\alpha_s$ of satellite orbit. Mean deviation of the east component of DOV is larger than that of the north (**a**) and vice versa (**b**).

For the results of TG2+J1J2+SA+C2, the mean deviation of the north component of DOV is 0.115 μrad, which is obviously smaller than the result of TG2 InIRA. The RMSE of the north component was 2.011 μrad, less than the results of TG2 InIRA and J1J2+SA+C2. The RMSE of the east component of DOV is 2.898 μrad, which is obviously smaller than the result of J1J2+SA+C2. And the mean deviation is -0.135 μrad, which has no significant change compared with that of TG2 InIRA and J1J2+SA+C2. This is to say the results of TG2+J1J2+SA+C2 significantly improved the mean deviation of the north DOV of TG2 InIRA and the RMSE of the east DOV of CAs. In addition, the RMSE of the north component

of J1J2+SA+C2 is smaller than that of TG2 InIRA, while the case of the east component is reversed. This is possibly the reason why the fused image of TG2+J1J2+SA+C2 is closer to that of J1J2+SA+C2 for the north component and to that of TG2 InIRA for the east component.

### 5.2. Comparison of Gravity Anomalies

Figure 16 shows the results of gravity anomalies from the data of TG2 InIRA, J1J2+SA+C2 and TG2+J1J2+SA+C2; as can be seen, they are highly consistent with each other. Figure 17 shows the corresponding results of the residual gravity anomalies. Compared with the result of J1J2+SA+C2, the TG2 InIRA image can exhibit finer textures and smaller fluctuations due to higher spatial resolution. As for the result of TG2+J1J2+SA+C2, the detailed information comes from the TG2 InIRA while it comes from J1J2+SA+C2 for the largely undulated areas, such as the central island chain and the seamount in the northeast.

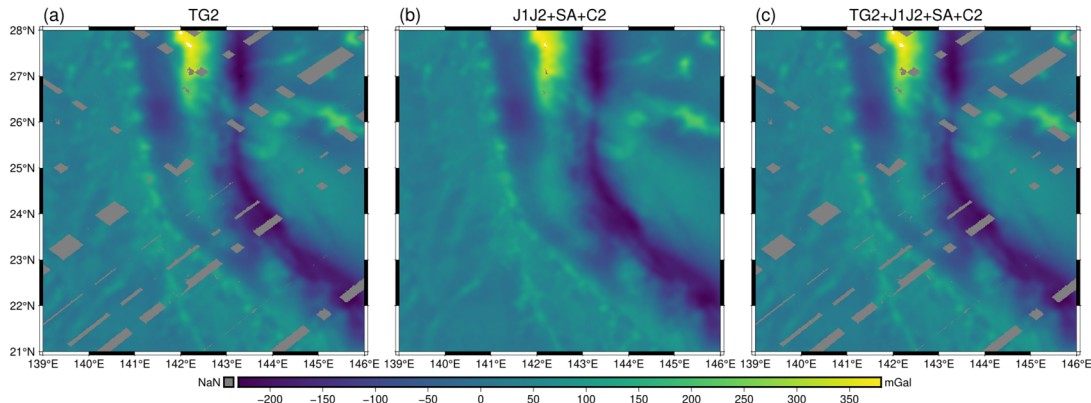

**Figure 16.** Gravity anomalies obtained by TG2 InIRA and CAs, (**a**) TG2, (**b**) J1J2+SA+C2, (**c**) TG2+J1J2+SA+C2.

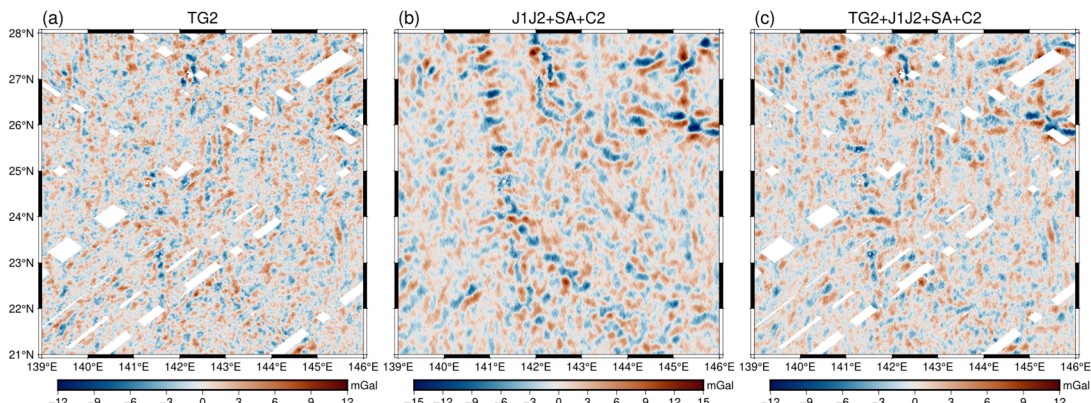

**Figure 17.** Residual gravity anomalies obtained by TG2 InIRA and CAs, (**a**) TG2, (**b**) J1J2+SA+C2, (**c**) TG2+J1J2+SA+C2.

In the following, we shall quantitatively evaluate the accuracies of recovered gravity anomalies by comparing with the published gravity models and shipborne gravity data. There are two widely recognized global marine gravity models. One is the SIO series model (e.g., V32.1) developed by Sandwell based on the fusion of multi-altimeter data [6], and the other one is the DTU series model (e.g., DTU17) developed by Andersen [74]. Table 5 lists the quantitative comparisons of the recovered gravity anomalies with the gravity models of SIO V32.1, DTU17 and EGM2008. The EGM2008 model is used in the remove–restore method. In order to evaluate the fused results more reasonably, an additional experiment is conducted using equal amount of CAs data and TG2 data. The fused SSHs of CAs (J1: 363 passes, J2: 304 passes, SA: 293 passes, C2: 293 passes) denoted as J1J2+SA+C2(equal).

**Table 5.** Comparison of the recovered gravity anomalies with gravity models.

| Unit: mGal | Max | Min | Mean | STD | RMSE |
|---|---|---|---|---|---|
| EGM2008 vs. TG2 | 12.121 | −12.702 | 0.003 | 2.060 | 2.060 |
| EGM2008 vs. J1J2+SA+C2 | 19.526 | −16.672 | 0.002 | 2.736 | 2.736 |
| EGM2008 vs. J1J2+SA+C2(equal) | 18.019 | −18.385 | 0.002 | 2.829 | 2.829 |
| EGM2008 vs. TG2+J1J2+SA+C2 | 13.209 | −11.906 | 0.001 | 2.033 | 2.033 |
| EGM2008 vs. TG2+J1J2+SA+C2(equal) | 12.907 | −10.656 | 0.001 | 1.959 | 1.959 |
| V32.1 vs. TG2 | 50.253 | −39.806 | 0.057 | 2.920 | 2.920 |
| V32.1 vs. J1J2+SA+C2 | 54.354 | −34.754 | 0.057 | 3.513 | 3.514 |
| V32.1 vs. J1J2+SA+C2(equal) | 51.482 | −31.914 | 0.056 | 3.580 | 3.580 |
| V32.1 vs. TG2+J1J2+SA+C2 | 51.330 | −40.186 | 0.056 | 2.974 | 2.975 |
| V32.1 vs. TG2+J1J2+SA+C2(equal) | 50.831 | −40.509 | 0.056 | 2.909 | 2.910 |
| DTU17 vs. TG2 | 14.038 | −18.681 | 0.054 | 2.218 | 2.218 |
| DTU17 vs. J1J2+SA+C2 | 29.810 | −16.482 | 0.053 | 2.835 | 2.835 |
| DTU17 vs. J1J2+SA+C2(equal) | 23.309 | −16.686 | 0.053 | 2.920 | 2.920 |
| DTU17 vs. TG2+J1J2+SA+C2 | 20.894 | −17.807 | 0.052 | 2.191 | 2.192 |
| DTU17 vs. TG2+J1J2+SA+C2(equal) | 16.492 | −18.155 | 0.052 | 2.124 | 2.125 |
| V32.1 vs. EGM2008 | 51.688 | −32.135 | 0.055 | 2.630 | 2.631 |
| DTU17 vs. EGM2008 | 18.802 | −12.253 | 0.051 | 1.423 | 1.424 |

As one can see from Table 5, the RMSEs of TG2 InIRA results compared with these three models are all about 0.6 mGal smaller than that of J1J2+SA+C2 results and J1J2+SA+C2(equal) results. This is to say the gravity recovery performance of TG2 InIRA is as good as, or even a little better, than that of CAs. For the TG2+J1J2+SA+C2 case, its RMSEs are slightly smaller than that of TG2 InIRA results compared with the EGM2008 and DTU models, but it is not when compared to the SIO model. As for the TG2+J1J2+SA+C2(equal) case, its RMSEs compared with the EGM2008, SIO and DTU models are all smaller than those of TG2 and J1J2+SA+C2(equal). The RMSE results of the fused gravities clearly show the role of improved east DOV of TG2 InIRA.

To evaluate the marine gravity anomalies by TG2 InIRA more extensively, we compare the results with the shipborne gravity measurements from the National Centers for Environmental Information (NCEI) as usually done when providing independent assessment. The shipborne gravimetry has the advantages of higher accuracy and higher resolution than the gravimetry developed by satellite altimeters [75]. Figure 18 presents the spatial distribution of the shipborne measured gravity anomalies by NCEI including about 57877 measurement data from 67 survey cruises. There are several long wavelength error sources affecting the shipborne gravity, such as gravimeter drift, absence of base-station ties, and uncertainty about the reference field used [76]. Therefore, preprocessing on shipborne gravity measurements is required before application. Generally, an EGM2008 model is adopted to unify the reference datum and the quadratic polynomial regression method is used to correct the long wavelength errors for each cruise [77]. First, the shipborne gravity measurements with differences compared to the EGM2008 gravity anomalies greater than 20 mGal are considered as outliers and excluded. Then, the correction model was used for each cruise [64]

$$\Delta g_{ship} = x_0 + x_1 \Delta t + x_2 \Delta t^2 \tag{16}$$

where $\Delta g_{ship}$ is the corrected value of the shipborne gravity measurement, $\Delta t$ is the time interval between the observation time and the starting time of the ship route; $x_0$, $x_1$ and $x_2$ are the coefficients obtained by using the least-squares method [20].

All the results of TG2, CAs and their fusion are interpolated and matched to the positions of the preprocessed shipborne measurements for comparison. The statistical results are listed in Table 6; as can be seen, the RMSE of TG2 InIRA results relative to the shipborne gravities is about 0.4 mGal smaller than those of J1J2+SA+C2 and J1J2+SA+C2(equal) results relative to the shipborne gravities. This is because the simultaneous wide swath measurement of TG2 InIRA helps for achieving balanced accuracies of the north and east components of the DOV, which is conductive to the gravity recovery.

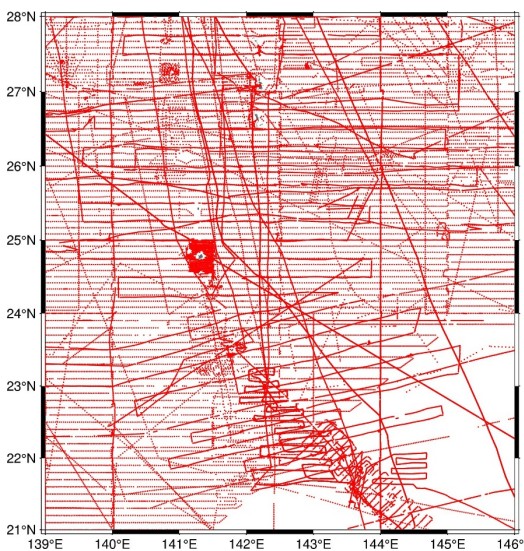

**Figure 18.** Spatial distribution of the NCEI shipborne gravity measurement data.

**Table 6.** Comparison of the recovered gravity anomalies with shipborne measurements.

| Unit: mGal | Max | Min | Mean | STD | RMSE |
|---|---|---|---|---|---|
| NCEI vs. TG2 | 27.142 | −27.563 | −0.209 | 4.992 | 4.997 |
| NCEI vs. J1J2+SA+C2 | 34.591 | −27.925 | −0.066 | 5.428 | 5.429 |
| NCEI vs. J1J2+SA+C2(equal) | 34.033 | −28.526 | −0.060 | 5.462 | 5.463 |
| NCEI vs. TG2+J1J2+SA+C2 | 26.623 | −25.921 | −0.153 | 5.016 | 5.019 |
| NCEI vs. TG2+J1J2+SA+C2(equal) | 26.029 | −26.160 | −0.167 | 4.964 | 4.967 |
| NCEI vs. V32.1 | 27.374 | −30.011 | −0.240 | 4.201 | 4.208 |
| NCEI vs. DTU17 | 21.794 | −23.987 | −0.223 | 4.334 | 4.340 |
| NCEI vs. EGM2008 | 23.761 | −23.155 | −0.195 | 4.792 | 4.796 |

The RMSE of TG2+J1J2+SA+C2(equal) result is all smaller than those of TG2 and J1J2+SA+C2(equal) compared with the shipborne gravities. The reason is that the higher accuracy of the north DOV of CAs improves the north DOV of TG2 InIRA, while the higher accuracy of the east DOV of TG2 InIRA improves the east DOV of CAs, leading to the final result superior to that of the single dataset. But as we have noticed, the TG2+J1J2+SA+C2 result is inferior to the TG2 InIRA result. And the possible reason is due to the worse east DOV of CAs with two times of TG2 InIRA data generates the major influence. These comparison results inspire us focus on how to design the fusion strategies and data volume allocation to preserve the advantages and weaken the disadvantages of different types of altimeters in the future.

In addition, the RMSEs of the published models relative to the shipborne measurements are all smaller than the above comparison results as can be expected. This is because the published models are developed using dedicated and sophisticated methods based upon the huge amount of satellite altimetry data lasting more than 20 years. It should be noted that the RMSEs of our recovered results compared with the NCEI measurements are slightly worse than that of EGM2008 because the noise cannot be effectively suppressed when the amount of data used is small or the recovery method is not dedicated enough.

According to the comparison results both with the published models and the shipborne gravity measurements, we can conclude that the gravity recovery accuracy of TG2 InIRA equals to that of CAs, and, in fact, the accuracy of TG2 InIRA is even slightly better. And by fusing the TG2 InIRA data with equal CAs data, the obtained gravity result can be significantly improved compared with the result by CAs alone and slightly improved compared with the result by TG2 alone.

## 6. Conclusions and Outlook

As the major approach for the measurement of short wavelength marine gravity field, the CAs have obvious limitations on measurement efficiency and spatial resolution due to the working principle of SSH measurement. Therefore, the WSA as a new generation altimeter has been paid to great attention for a long time and is expected to overcome these limitations and improve the measurement accuracy.

In this paper, we use the SSH data by TG2 InIRA to conduct the gravity recovery experiment in the Western Pacific region. The results show the feasibility of wide-swath SSH measurement and demonstrate the advantages of TG2 InIRA in gravity recovery not only in the measurement efficiency, but also in the achieved accuracy benefited from the realized consistency between the north and east components of DOV. Compared with the published gravity models, the RMSEs of TG2 InIRA are all about 0.6 mGal smaller than that of CAs, while compared with the shipborne data, the result of TG2 InIRA is about 0.4 mGal smaller than that of CAs. More importantly, the fused gravity result using equal TG2 InIRA data and CAs data performs better than that using TG2 InIRA data alone and CAs data alone.

We should emphasize that the achieved performance of TG2 InIRA is just based on a limited system design, e.g., the signal bandwidth is only 40 MHz, and just one side of the track is looked (i.e., single wide-swath). As we know, the larger the signal bandwidth, the higher the accuracy of range (or SSH) measurement for CAs, and that is why almost all the operational CAs adopt wide-band signal of at least 320 MHz bandwidth. It is also true for WSA under off-nadir geometry because the range resolution can be higher by using larger bandwidth signal and correspondently the independent samples can be increased covering the same swath, which is good for reducing the random error of interferometric phase measurement by just averaging the samples and, thus, the SSH accuracy can be enhanced. The already launched SWOT altimeter has more advanced measurement capabilities [34], as shown in Table 7. The double wide-swath feature can not only improve the observation efficiency, but also facilitate the error corrections including the above mentioned BRE and BLE corrections. If operational WSA observation data are used in the future, the marine gravity anomalies can be measured more accurate than ever before.

**Table 7.** Parameter comparison of TG2 InIRA and SWOT KaRIn.

| Parameter | TG2 InIRA | SWOT KaRIn |
|---|---|---|
| Orbital altitude | 390 km | 890 km |
| Inclination | 43° | 77.6° |
| Center frequency | 13.58 GHz | 35.75 GHz |
| Bandwidth | 40 MHz | 200 MHz |
| Baseline length | 2.3 m | 10 m |
| Baseline inclination | 5° | 0° |
| Swath width * | 35 km | 120 km |
| Swath averaged height error * | 5.7 cm@5 km $\times$ 5 km | 2.4 cm@1 km $\times$ 1 km |
| Spatial resolution (across-track) * | 30–200 m | 10–70 m |
| Spatial resolution (along-track) * | 30 m | 2.5 m |

* These parameters for TG2 InIRA are practically achieved, while those for SWOT KaRIn are theoretically calculated.

Last but not least, we should care about the spatial resolution issue. The spatial resolution of DOV and gravity anomalies by TG2 InIRA has been visually shown higher than that by CAs. This may be helpful for improving the spatial resolution of seafloor topography and plate tectonics. Thus, it deserves us to further explore the high spatial resolution capability of InIRA in the future by using the TG2 InIRA SSH data sampled at 1 km $\times$ 1 km or even 500 m $\times$ 500 m grids.

**Author Contributions:** Conceptualization, Y.Z. and M.S.; methodology, M.S., X.D. and Y.Z.; software, M.S.; validation, M.S.; formal analysis, M.S. and Y.Z.; writing—original draft preparation, M.S.; writing—review and editing, Y.Z., M.S., X.D. and X.S.; visualization, M.S. All authors have read and agreed to the published version of the manuscript.

**Funding:** This research was funded by the National Key R&D Program of China, grant number 2016YFC1401004 along with the China Manned Space Program.

**Data Availability Statement:** TG2 InIRA SSH data are managed by the Space Science and Application Data Service Platform for China Manned Space Engineering (http://www.msadc.cn/main/home, accessed on 10 September 2019) which is currently only accessible to specific users. The SSH data of J1, J2 and SA and MSS model can be downloaded from AVISO site, and the SSH data of C2 can be downloaded from ESA site. The shipborne gravity data can be downloaded from NCEI (https://www.ncei.noaa.gov, accessed on 5 September 2021). The DTU models can be downloaded from DTU site (https://ftp.space.dtu.dk/pub, accessed on 5 July 2022), and the SIO models can be downloaded from SIO site (https://topex.ucsd.edu/pub, accessed on 5 July 2022). The EGM2008 model can be downloaded from ICGEM (http://icgem.gfz-potsdam.de, accessed on 5 November 2021).

**Acknowledgments:** The authors thank the AVISO and ESA for altimeter data and the NCEI for shipborne data, the DTU and SIO for model data, and the open-source software GMT for making the figures. We also would like to thank all the editors and reviewers of this paper for their constructive feedback, which helps us improve the readability and completeness of the paper.

**Conflicts of Interest:** The authors declare no conflict of interest.

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
