# Peer review of "Preliminary Results of Marine Gravity Recovery by Tiangong-2 Interferometric Imaging Radar Altimeter"

_remotesensing, doi:10.3390/rs15194759_

Round 1

Reviewer 1 Report

This study presents the results of ocean gravity anomalies recovery using data from the interferometric imaging radar altimeter carried by the TianGong-2 space laboratory. TG2, as the first space laboratory equipped with an interferometric imaging radar altimeter, enables two-dimensional wide-swath measurements of sea surface heights. The data volume and observation efficiency of TG2 are higher than those of satellites equipped with nadir altimeters. The article provides experience for future exploration in the field of interferometric radar altimetry. I would like to suggest publication of the manuscript after the following revisions.

Main comments:

L83: Please kindly provide the in-orbit operational duration of TianGong-2, such as from xx Time to xx Time, in-orbit observations for about 27 months. Additionally, could you offer a brief overview of the cycle and pass quantities for TG2, and how many repeated observations it can cover a specific region during its orbital tenure?  

L101: “Two times of TG2 InIRA data from CAs are also collocated.” The description in a single sentence might be somewhat unclear. I suggest breaking it into two separate clauses. For example, "The data volume of CAs is twice that of TG2 InIRA, and this data has also been collected."  

L110: Please clearly indicate the frequency band in which TG2 operates, such as Ku-band.

L141: In this paragraph, only “First” expresses what the first step did, not “Then”, or the “Final” step.

L149: Please change “The interferometric phases…is” to “The interferometric phase…is”.

L153: "sea surface" appears several times in the text, missing the definite article "the" 

L157: Please change “…and thus two true ranges can be obtained with one of which decided by the time delay” to “…and thus two true ranges can be obtained, one of which is decided by the time delay.”

L188: Does ionospheric/Tropospheric delay belong to geophysical errors?  

L205: Please change “…waveform, but…” to “…waveform. But…”.

L272: “71 passes…”, please give more information such as the time, how many cycles these passes are distributed in, etc. 

L325: The original resolutions of SSH obtained by TG2 InIRA are about 30 m along-track and 30-200 m across-track. The two “original grids” in Figure 7 (a) point to the same grid, maybe the image needs to change: one point to the along-track grid and another point to the across-track grid.

L477: This sentence is too long, you can split it.

L488: Please change “…with finer textures exhibited resulted from…” to “…with finer textures exhibited resulting from…”.

L576: This article only compares results with the SIO V32.1 and DTU17 models, and does not need to mention SIO V27.1 and DTU13.

Line 600: According to Table 4, TG2 performs best than other results except the fused results including TG2 data. However, in Table 3, the vertical deflections do not perform best. Especially error of TG2 is the largest in terms of error mean in Table 3, but not in Table 4. The author should check the results and explain the reason on this point.

Reviewer 2 Report

I have been reading the manuscript “Preliminary results of marine gravity recovery by Tiangong-2 interferometric imaging Radar Altimeter” and I find the paper both interesting and well written. I suspect that the paper have been submitted to another journal and transferred to Remote sensing since it seems to be well revised already.

I have a few minor concerns figures.  

1)    The quality of the pictures. I think that something went wrong here. Figure 2+3 are hard to read and this goes with several other figures as well.

2)    I would add a figure 6c with bathymetry of the region or setting of the region of investigation.

Secondly I suggest  a few clarifications.

Line 175. I am a bit curious why multi-looking should decrease the resolution. It normally enhances it. I would rewrite this sentence.

Line 299/300 and figure text for figure 6.

You use the word “times”. This is not correct English. Number of Cycles or repeats would be much better.

Line 307. You remove SSH anomalies greater than 5 meters and this removes 0.06%. This is a VERY high limit and normal 2 or 3 meters are used. You can maintain but perhaps write that its not critical which limit is chosen.

Section starting at line 312

You resample the data into 2x2 km. How does this influence your results. I would think that averaging would give a more solid approach to this.

Figure 8. This reveals that there are some spatial gaps in the TG2 data?. Have these gaps been taken into account  in the following intercomparison with tide gauges in Table 4/5. I feel that this could affect the results if you left in-situ data in these regions out of the comparisons?.

I liked your explanation of the BRE and BLE are correction specifically to Tiangong? I am asking as the paper would

stand stronger if you added some preliminary comments related to how this mission is similar/differ to Surface Water and Ocean Topography mission which is now successfully launched. Perhaps this would be an improvement to the conclusion.

Reviewer 3 Report

This paper entitled “Preliminary results of marine gravity recovery by Tiangong-2 Interferometric Imaging Radar Altimeter” mainly studied the TG2 InIRA data processing and then converted the DOV to gravity anomalies. It is a very interesting study. However, I have to say this manuscript does not meet the standards of a high-quality scientific paper right now. The theory and processing strategy are not new, and the conclusions are not very well supported by the numerical results. On the other hand, the method described is very simple and not clear in some places, and readers obviously can not understand the content based on the current draft, e.g., not clear how to choose many parameters and the corresponding influence. At last, there is no open access to the TG2 InIRA data and software. Therefore, I suggest a major revision and hope the authors can improve the paper significantly.

1, Abstract: Please brief it and remove unnecessary information.

2, Some figures should be much improved, e.g., 2, 3, 5 and so on.

3, Add some flowcharts into the methodology part and make it clearer. It is a little bit messy in the current version. 

4, Add some statistics for some figures, e.g., Fig. 12, 13, and so on, to make it more comparable. 

5, Use an updated gravity field model, e.g., EIGEN6C4 instead of EGM2008. Some studies have already found quite significant differences between the latest global gravity models and EGM2008, e.g.,

Lu, B., Xu, C., Li, J., Zhong, B. and van der Meijde, M., 2022. Marine gravimetry and its improvements to seafloor topography estimation in the southwestern coastal area of the Baltic Sea. Remote Sensing, 14(16), p.3921.

Förste, C. et al. EIGEN-6C4 The latest combined global gravity field model including GOCE data up to degree and order 2190 of GFZ Potsdam and GRGS Toulouse (GFZ Data Services, 2014); http://doi.org/10.5880/icgem.2015.1

Zingerle P, Pail R, Gruber T, et al. The combined global gravity field model XGM2019e[J]. Journal of geodesy, 2020, 94: 1-12.

6, I really doubt the processing strategy for the shipborne gravity measurements, E.q. 16. What are the accuracy and spatial resolutions here? Please try to find some better air-marine gravimetry data to check your numerical results.

7, From Tables 4 and 5, and the statement in the paper, TG2 seems better than traditional satellite altimetry, but when compared to DTU17 or V32.1, TG2 is obviously worse. The authors mentioned the reason in Lines 637-639: “This is because …”. I think it is not enough here. Please explain why TG2 is better than J1J2, SA, C2, or a combination, but worse than the models. Either should change the statements or make clear the reason.

8, Please add some comparisons to SWOT.

9, Why is equal “fusion” better than single datasets? “Equal” combination may not be enough for different datasets with different accuracy and spatial resolution/distribution!

10, Conclusion, the improvement of 0.6 or 0.4 mGal is not that significant, and in the condition that the accuracy and spatial resolution of the shipborne gravimetry is not clear. Please remove or rewrite statements like, we believe …, we confident … and so on. 

11, Please improve the language and format.

11, Please improve the language and format.

Round 2

Reviewer 3 Report

The authors have improved the manuscript significantly. It is worth publishing these preliminary results. On the other hand, there are still many remaining key questions that need/should be studied in the future. 

Minor editing of English language required.
